# Potential Applications of Native Cyanobacterium Isolate (*Arthrospira platensis* NIOF17/003) for Biodiesel Production and Utilization of Its Byproduct in Marine Rotifer (*Brachionus plicatilis*) Production

Mohamed A. Zaki [1], Mohamed Ashour [2,*], Ahmed M. M. Heneash [2], Mohamed M. Mabrouk [3], Ahmed E. Alprol [2], Hanan M. Khairy [2], Abdelaziz M. Nour [1], Abdallah Tageldein Mansour [4,5], Hesham A. Hassanien [4,6], Ahmed Gaber [7] and Mostafa E. Elshobary [8,9,*]

1. Animal and Fish Production Department, Faculty of Agriculture, Alexandria University, Alexandria 21545, Egypt; mohamed_zaki4@yahoo.com (M.A.Z.); nouraziz2000@yahoo.com (A.M.N.)
2. National Institute of Oceanography and Fisheries (NIOF), Cairo 11516, Egypt; aheneash@yahoo.com (A.M.M.H.); ah831992@gmail.com (A.E.A.); hanan_khairy@yahoo.com (H.M.K.)
3. Fish Division, Animal Production Department, Faculty of Agriculture, Al-Azhar University, Cairo 11823, Egypt; mabrouk3m@yahoo.com
4. Animal and Fish Production Department, College of Agricultural and Food Sciences, King Faisal University, P.O. Box 420, Al-Ahsa 31982, Saudi Arabia; amansour@kfu.edu.sa (A.T.M.); helsanwey@kfu.edu.sa (H.A.H.)
5. Fish and Animal Production Department, Faculty of Agriculture (Saba Basha), Alexandria University, Alexandria 21531, Egypt
6. Animal Production Department, Faculty of Agriculture, Cairo University, Gamma St., Giza 12613, Egypt
7. Department of Biology, College of Science, Taif University, P.O. Box 11099, Taif 21944, Saudi Arabia; a.gaber@tu.edu.sa
8. Department of Botany and Microbiology, Faculty of Science, Tanta University, Tanta 31527, Egypt
9. School of Food & Biological Engineering, Jiangsu University, Zhenjiang 212013, China
* Correspondence: microalgae_egypt@yahoo.com (M.A.); mostafa_elshobary@science.tanta.edu.eg (M.E.E.); Tel.: +20-1283184088 (M.A.)

**Abstract:** To achieve strong, successful and commercial aqua-biotechnological microalgae applications, screening, isolation, molecular identification, and physiological characterizations are needed. In the current study, a native cyanobacteria strain *Arthrospira platensis* NIOF17/003 was isolated from the surface water of El-Khadra Lake, a saline-alkaline lake located in Wadi El-Natrun, Egypt. The cyanobacterium was phylogenetically identified by 16S rRNA molecular marker and deposited in the GenBank database (accession number MW396472). The late exponential phase of *A. platensis* NIOF17/003 was reached at the 8th day of growth using Zarrouk medium, with a recorded dry weight (DW) of 0.845 g L$^{-1}$. The isolated strain showed 52% of protein, 14% of carbohydrate, biomass productivity of 143.83 mg L$^{-1}$ day$^{-1}$, 8.5% of lipid, and lipid productivity of 14.37 mg L$^{-1}$ day$^{-1}$. In general, the values of cetane number, iodine value, cold filter plugging point (52.9, 85.5 g I2/100 g oil, and $-2.2$ °C, respectively) of the isolated fatty acid methyl esters are in accordance with those suggested by international standards. Besides, applying algal-free lipid (FL) as biodiesel byproduct in the production of rotifer (*Brachionus plicatilis*) revealed that a 0.6 g L$^{-1}$ FL significantly increased the rotifer population females carrying eggs, confirming that FL can be used efficiently for *B. plicatilis* production. The current study concluded that the new isolate *A. platensis* NIOF17/003 is a promising strain for double sustainable use in biodiesel production and aquaculture feed.

**Keywords:** *Arthrospira platensis* NIOF17/003; aquaculture; biodiesel production; byproduct; live-feeds; blue-green algae; El-Khadra Lake; alkaline lake; Wadi El-Natrun

## 1. Introduction

Rapid consumption of fossil fuel requires the substitution-sustainable fuel source, which can substitute the traditional fuel for the interpolation of the energy crisis with a

lower environmental effect. Researchers and scientists are working to discover renewable, sustainable, and eco-friendly energy sources to replace or reduce the conventional fuel excess load. Therefore, discovering a sustainable source of fuel is a constantly dynamic issue [1]. Nowadays, biodiesel becomes an appropriate substitution tool for researchers and scientists for supplementing conventional fuel. Biodiesel's characteristics are very close to fossil diesel that can be used in the existing engine without any modulation. At present, biodiesel's production cost is from 1.5 to 3-fold more than fossil diesel due to the high cost of raw feedstocks and unavailability of oil crops utilized for human food consumption [2].

Algal biomass are considered an attractive natural source of bioactive materials that can be utilized in several applications such as food industry [3], pharmaceuticals [4], cosmetics [5], biogas and biodiesel [6,7], antimicrobial and antioxidant activities [3,8], aquaculture [9–12], biofertilizers and bioremediation [13–16]. Therefore, the screening of native microalgae species is necessary to realize a potent algal database with effective commercial biotechnological applications [3,17]. In the near future, it is presaged that algal biomass will be the most sustainable source for biodiesel production, due to its biodiesel-oil contents which can reach more than 40-fold higher than those of other plants [18]. In addition, the algal-free lipid biomass is filled with several other components that can be utilized in many applications, including aquaculture feeds, animal feed additives, and valuable bioactive compounds for cosmetic and pharmaceutical products [17,19].

Cyanobacteria, filamentous blue-green algae, have wide ranges of biotechnological applications [2,4,14]. Over the world, *Arthrospira* (formerly named *Spirulina*) is one of the most famous cyanobacterium strains that has been extensively cultured at mass commercial scale [20,21]. The global production of *Spirulina* grew from 48 thousand tons in 2005 to over 89 thousand tons in 2016 [22]. Until now, more than 30 species of *Arthrospira* have been identified. The well-known dominant species includes *Arthrospira platensis*, *A. maxima*, *A. major*, *A. laxissima*, *A. caldaria*, *A. princes*, *A. subtilissima*, *A. subsalsa*, *A. spirulinoides* and *A. curta* [23]. Due to the unique metabolic compounds and biochemical composition of *A. platensis*, this species is a strong candidate to utilize in many biotechnological applications such as human food supplements, pharmaceuticals, animal feed additives, omega- 3 fatty acids, essential amino acids, biofertilizer, and pigments [4,14,21,23,24]. Recently, due to the considerations of biomass and lipid productivity, many authors stated that *A. platensis* is considered one of the most magician feedstock for biodiesel production [2,20,23,25–27]. Interestingly, several studies reported that the quality of biodiesel extracted from *Arthrospira*-oil is agreed with the recommended specifications of international standards of Europe (EN 14214) and USA (ASTM6751−03) [23–27].

*A. platensis* species has the ability to exist and grow in various types of water like seawater [28], freshwater [29], aquaculture wastewater [30], industrial wastewater [31], agriculture wastewater [32], and domestic wastewater [33]. *A. platensis* is a filamentous, spiral-shaped, blue-green alga that culture in alkaline water, with pH up to 11 [20]. These unique environmental conditions have the priority to prevent external contamination by pathogenic bacteria and/or other families of microalgae, while it cannot prevent the contamination by other blue-green algae. *A. platensis* is not easily influenced by changes in physical and chemical environmental conditions, like nutrient limitations, water temperature, light intensity, and salinity. *Arthrospira* strains naturally exist in high alkaline water which contains carbonate or bicarbonate [23]. In Egypt, El-Khadra Lake is an Egyptian shallow alkaline lake located in Wadi El-Natrun, a depression hydrostatically jointed with Delta and its lakes and hosts figures of main shallow alkaline lakes, reaching area about 162 acres. The weather at Wadi El-Natrun is arid and naked, which leads to continuous evaporation of its lake water generating great crusts of salts [34]. Determination of cyanobacteria in El-Khadra Lake, especially *Spirulina* species which are a dominant species in the El-Khadra Lake, had already been reported [34–38]. However, *Arthrospira* species of El-Khadra Lake is not yet exposed to evaluate their potential as biodiesel or aquaculture feedstuff.

Therefore, the current work aims to study the potential of biodiesel production, in integration with aquaculture feedstuff as biodiesel-byproduct, of *Arthrospira* strain, a native pure strain isolated from the El-Khadra Lake. The phylogenetic identification was carried out using 16S rRNA molecular marker. The growth, biochemical composition, lipid and biomass productivity of the isolated strain were determined. In addition, the specification of biodiesel produced from the oil of isolated strain was evaluated and compared with international standards and with other recommended strains. Moreover, the whole biomass (WB) of isolated strain and the residual algal-free lipid biomass (FL) after oil extraction were evaluated for their potential utilization as the primary feedstock of marine rotifer (*Brachionus plicatilis*).

## 2. Materials and Methods

### 2.1. Sampling

Water samples were collected in 2017 spring season, at noon, from the water surface of El-Khadra Lake (30°26′504″ N; 30°13′546″ E), Wadi El-Natrun, Egypt (Figure 1). Water samples were possessed in sterilized bottles and transferred shortly after collection to the Microalgae Room, Invertebrates Laboratory, National Institute of Oceanography and Fisheries (NIOF), Alexandria, Egypt. In the field site, water quality parameters of temperature, pH, and salinity were measured using a pH/temperature meter (Milwaukee MW102, Milwaukee Instruments, Inc., Rocky Mount, NC, USA) and a portable conductivity meter (Oakton, Eutech Instruments, Vernon Hills, IL, USA). $CO_3$, $HCO_3$, electrical conductivity (EC), and total soluble salts (TSS) values of the water surface of El-Khadra Lake were determined according to APHA [39].

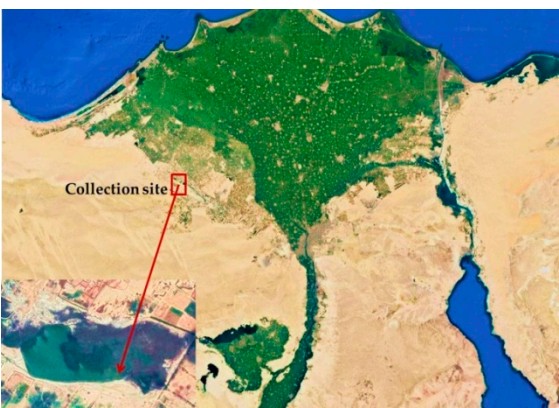

**Figure 1.** Location of the cyanobacterium collection site (El-Khadra Lake 30°26′504″ N; 30°13′546″ E, Wadi El-Natrun, Egypt).

### 2.2. Isolation and Purification

Cyanobacterium isolate was purified and conducted on Zarrouk medium [40] in agar solid medium followed by serial dilution method as described previously [41]. Cyanobacterium isolate was initially investigated by a microscope (Olympus, light microscope, INV-100, Esse3, MB, Italy). The purified isolate was kept and sub-cultured in 500 mL of sterilized Zarrouk medium and incubated under controlled temperature conditions (28.5 ± 1.5 °C), illumination (3500–4500 Lux day$^{-1}$), and continuous aeration with shaking at 80 rpm. The pure isolated strain was morphologically identified under light microscope as *Arthrospira* (*Spirulina*) species, using the following key references [42–44].

### 2.3. Phylogenetic Identification

The total genomic DNA was extracted from dried *Arthrospira* powder following the standard CTAB (cetyltrimethylammonium bromide) protocol modified from Grube et al. (1995) [45] with minor modifications [46]. The 16S rRNA gene was amplified using the universal primers, 27F (5′-AGAGTTTGATCCTGGCTCAG-3′) and 1492R (5′-

TACGGCTACCTTGTTACGACTT-3′) with Super Pfx DNA Polymerase (Life Technologies, Grand Island, NY, USA). PCR thermal cycle initiated with denaturation of DNA at 98 °C for 3 min, followed by 34 cycles with denaturation at 98 °C for 10 s, annealing at 58 °C for 30 s, elongation at 72 °C for 25 s, and final elongation was performed at 72 °C for 10 min. PCR products were visualized in 1% (*w/v*) agarose gel electrophoresis. DNA fragments that corresponded to the particular size of 16S rRNA gene were purified by the FastPure Gel DNA Extraction Mini Kit and later sequenced in Hongxun Technology (Suzhou, China). The purified 16S rDNA sequences were aligned and compared with different 16S rDNA sequences using the nBLAST search (http://www.ncbi.nlm.nih.gov/blast) to find the closely related species. The sequences were aligned using MUSCLE with the default parameters (MEGA X software) (www.megasoftware.net). A dendrogram was created using the neighbor-joining (NJ) algorithm based on the parameter distance (PD) using MEGA X software [47].

### 2.4. Growth, Biomass, and Biochemical Constituent

The cyanobacterium isolate was cultivated for 12 days in a batch culture system using 1 L conical flask filled with 500 mL of sterilized Zarrouk medium. Batch cultures were maintained under controlled conditions of temperature ($28.5 \pm 1.5$ °C), continuous illumination ($3500$–$4500$ Lux day$^{-1}$) by tubular fluorescent lamps (PHILIPS Master TL-D 85 W/840), and continuous aeration (3 cm$^3$ min$^{-1}$) was continuously applied to the cultures by a sterile filtered air, with shaking at 80 rpm. The isolate growth was conducted by the determination of optical density (OD$_{560}$), growth rate, dry weight (DW), and biochemical composition at the late exponential phase (LEP, day 8th). OD was monitored using a spectrophotometer (CECIL 2010, double beam) as a standard curve by measuring the initial optical density at (OD$_{560}$) of $\approx 0.05$ [48]. Biomass productivity was evaluated according to Abomohra et al. [49] as the following equation:

$$\text{Biomass productivity (mg L}^{-1}\text{ day}^{-1}) = (\text{DW}_L - \text{DW}_E) \times (t_L - t_E)^{-1} \tag{1}$$

where, DW$_E$ and DW$_L$ are the DW (mg L$^{-1}$) when the growth reaches to the days of early exponential phase (t$_E$) and the days of late exponential phase (t$_L$), respectively.

For biochemical analysis, once the culture accomplished to late exponential phase (LEP, day 8th), 10 mL of culture water was centrifuged at $7000\times g$ for 10 min. The supernatant was neglected, and the pellet was preserved at $-20$ °C for further analysis. Extraction and determination of total protein were conducted according to Rausch's methods [50] and Hartree [51], respectively. Extraction and determination of total carbohydrate were conducted according to Myklestad and Haug [52] and DuBois et al. [53], respectively. Extraction and determination of total lipid were performed according to the method previously described by Elshobary et al. [54]. After oil extraction, the residual algal cells, called algal-free lipid (FL), were dried (55 °C for 48 h) and preserved ($-20$ °C) for further analyses. The extracted oil was subjected to transmethylation using sodium methoxide methods as previously described by Ashour et al. [17] for GC analysis. Briefly, 50 mg of dried lipid extracts were mixed with 666 μL of methanol: toluene (50:50, *v/v*) and 333 μL of 0.5 M sodium methoxide were added and incubated for 20 min at room temperature. After incubation, 1 mL of NaCl (1 M) and 100 μL of HCl (37%) were added, then FAMEs were extracted by addition of 3 mL of hexane. The upper hexane phase was transferred to a clean tube for evaporation. FAMEs were redissolved in 20 μL of acetonitrile and subjected to GC analysis. Lipid productivity was calculated as the method described by Wen et al. [55] using the following equation:

$$P_{Lipid} \text{ (mg L}^{-1}\text{ day}^{-1}) = C_{Lipid} \text{ (g g}^{-1}) \times \text{DW (g L}^{-1})/\text{T (day)} \tag{2}$$

where $P_{Lipid}$ and $C_{Lipid}$ is the lipid productivity and lipid contents of algal cells, respectively, DW is dry weight of algal cells, and T is the time of cultivation.

### 2.5. Determination of Biodiesel Properties

The physicochemical characterizations of FAMEs of the biodiesel produced from cyanobacterium were calculated.

Unsaturation degree (DU, %), saponification value (SV, mg KOH g$^{-1}$), iodine value (IV, g I2 100 g$^{-1}$ oil), cetane number (CN), cold filter plugging point (CFPP, °C), long-chain saturation factor (LCSF, wt%), and kinematic viscosity (KV, mm$^2$ s$^{-1}$), were determined as described by Ramirez-Verduzco et al. [56] and Ashour et al. [17], as the following equations:

$$DU = \Sigma \, MUFA + (2 \times PUFA) \tag{3}$$

$$SV = \Sigma \, [(560 \times N\%)/M] \tag{4}$$

$$IV = \Sigma \, [(254 \times N\% \times D)/M] \tag{5}$$

$$CN = 46.3 + (5458/SV) - (0.225 \times IV) \tag{6}$$

$$KV = -12.503 + (2.496 \times \ln M) - (0.178 \times D) \tag{7}$$

$$LCSF = (0.1 \times C16{:}0) + (0.5 \times C18{:}0) + (1 \times C20{:}0) + (1.5 \times C22{:}0) + (2 \times C24{:}0) \tag{8}$$

$$CFPP = (3.1417 \times LCSF) - 16.477 \tag{9}$$

where, N% is the percentage of each fatty acid; MUFA and PUFA are the percentage of monounsaturated and polyunsaturated fatty acids, respectively; M is the molecular mass of each fatty acid; D is the number of carbon-carbon double bonds; and C16:0, C18:0, C20:0, C22:0 and C24:0 are the percentages of the corresponding fatty acids.

### 2.6. Application of Algal-Free Lipid as Aquaculture Feed

After biodiesel extraction, the algal-free lipid (FL), in comparison with whole biomass before lipid extraction (WB), was evaluated as feed for marine rotifer *Brachionus plicatilis* (L-type with average length about 180 μm). Before the experiment, *B. plicatilis* was cultured under controlled conditions of temperature (27 ± 1 °C), salinity (30 ± 1 ppt), pH (7.7 ± 0.15), continuous gentle aeration, and enriched with native isolate marine microalga *Nannochloropsis oceanica* NIOF15/001, previously isolated and identified by [17], at density $5 \times 10^6$ cells mL$^{-1}$ day$^{-1}$. *B. plicatilis* were harvested from the stock culture tanks and transferred to the new culture water for a 24 h gut evacuation. After 24 h of starvation, *B. plicatilis* were distributed with a constant initial stock density of 20,000 individual L$^{-1}$ of filtered seawater with different concentrations (0.2, 0.4, 0.6, 0.8, 1.0 g FL or g WB) of FL (FL$_{0.2}$, FL$_{0.4}$, FL$_{0.6}$, FL$_{0.8}$, FL$_{1.0}$, respectively) and WB (WB$_{0.2}$, WB$_{0.4}$, WB$_{0.6}$, WB$_{0.8}$, WB$_{1.0}$, respectively). Each concentration (treatment) was conducted in three replicates. The rotifer *B. plicatilis* feeding trial was continued for 72 h in batch culture system. The experiment was monitored under controlled conditions of (22 ± 1 °C), salinity (30 ± 1 ppt), and pH (7.7 ± 0.15), without aeration. Rotifer growth indices were evaluated based on population (increase in number) and the number of females carrying eggs. The counts of rotifer individuals were determined using a Sedgwick-Rafter counting cell, under an optical microscope. Rotifer individuals were transferred to glass slides covered with coverslips to quantitative sample counts.

### 2.7. Statistical Analysis

Data were presented (*n* = 3) as mean ± standard deviation (SD). Before analysis, normality and homoscedasticity suspicions were affirmed. Statistical analyses for results were conducted using SPSS (IBM, v. 20, Armonk, NY, USA). All evaluated variables were conducted to an analysis of variance (ANOVA) followed by Duncan's multiple range tests and least significant difference (LSD) test, at a significant level (*p* < 0.05).

## 3. Results

### 3.1. Water Sample Characterizations

Surface water characteristics of El-Khadra Lake are shown in Table 1.

**Table 1.** Physicochemical characteristics of water surface of El-Khadra Lake.

| Characteristics | Values |
|---|---|
| Temperature (°C) | 27.50 ± 0.75 |
| pH | 9.36 ± 0.17 |
| $CO_3$ (ppt) | 2.80 ± 0.18 |
| $HCO_3$ (ppt) | 8.55 ± 260 |
| Conductivity (EC, ds/m) | 87.70 ± 1.90 |
| Total Soluble Salts (TSS, ppt) | 67.30 ± 1.10 |

Presented data were as mean ± standard deviation ($n = 3$).

### 3.2. Phylogenetic Identification

The 16S rDNA gene phylogeny was inferred from more than 1383 bp nucleotide sequences (PCR-based). From the maximum likelihood phylogenetic tree based on the 16S rRNA gene, the 16S gene region was aligned, with 16S nucleotide sequences of 12 *Arthrospira* strains in the NCBI Ribosomal DNA sequence plus three 16S rDNA sequences of *Lyngbya* and *Neolyngbya* sp. as an outgroup in the NCBI rDNA sequence. Each species formed monophyletic clades and the isolated strain was grouped in *Arthrospira platensis* (KC195868, KC536647, and KM019966) with high similarity reached to 100%, 97.5% and 85% bootstrap, respectively (Figure 2). The newly isolated *Arthrospira platensis* NIOF17/003 sequence was deposited in the GenBank database (accession number MW396472).

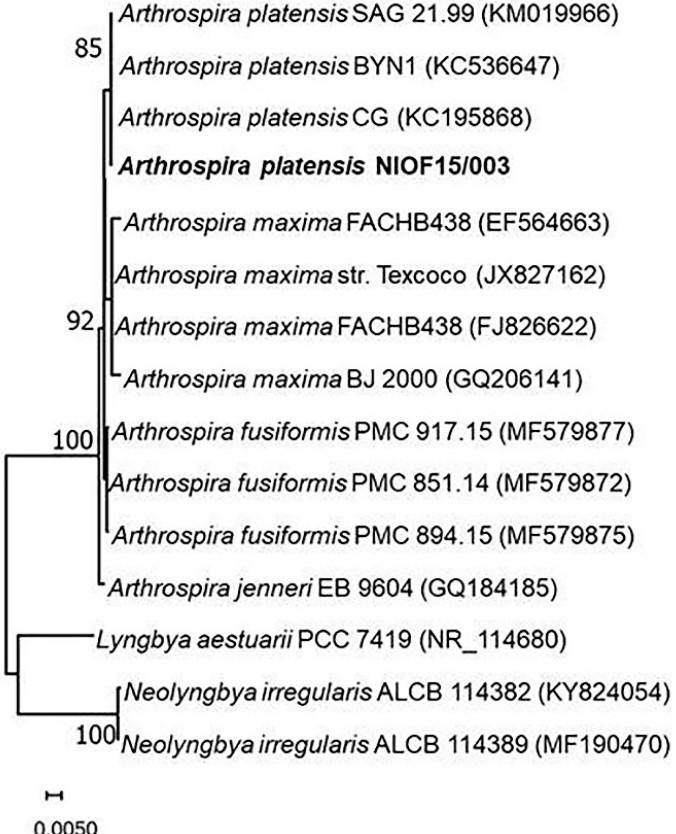

**Figure 2.** Neighbor-joining (NJ) phylogenetic tree for *Arthrospira platensis* NIOF17/003 (accession number MW396472) based on using 16S rRNA sequences. Bootstrap values > 70 are shown in order of NJ distance.

### 3.3. Growth and Biochemical Composition

Figure 3 shows the growth phases of *A. platensis* NIOF17/003 cultured on Zarrouk medium for 12 days; namely, early and late exponential phases (EEP and LEP, respectively) and early stationary phase (ESP). Table 2 shows the dry weight (0.845 g $L^{-1}$), proteins (52.03%), lipids (8.52%), and carbohydrates (14%), based on cell dry weight, at LEP. Biomass and lipid productivity, at LEP, was 143.83 and 14.37 mg $L^{-1}$ $day^{-1}$, respectively, as presented in Table 2.

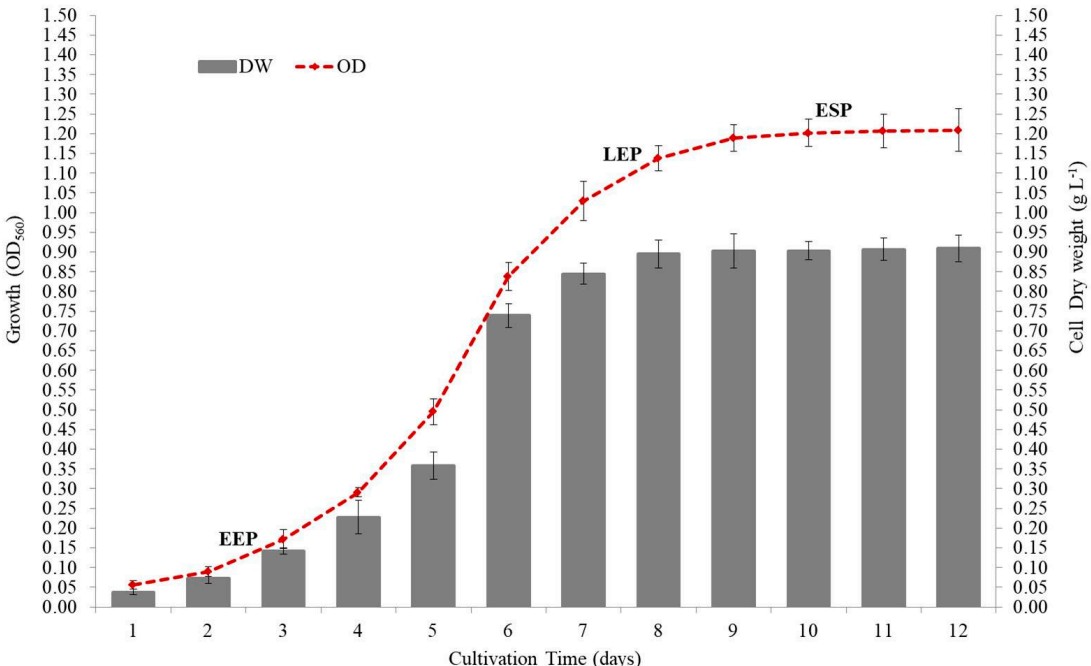

**Figure 3.** Dry weights of *Arthrospira platensis* NIOF17/003 at different growth rate phases; early exponential phase, late exponential phase, and early stationary phase (EEP, LEP and ESP, respectively).

**Table 2.** Biochemical composition, growth and productivity of *Arthrospira platensis* NIOF17/003.

| Parameter | Values |
|:---:|:---:|
| Dry Weight (DW, g $L^{-1}$) | $0.845 \pm 0.038$ |
| Proteins (% DW) | $52.03 \pm 0.77$ |
| Lipids (% DW) | $8.52 \pm 1.04$ |
| Carbohydrates (% DW) | $14.00 \pm 1.67$ |
| Biomass productivity (mg $L^{-1}$ $day^{-1}$) | $143.83 \pm 3.69$ |
| Lipid productivity (mg $L^{-1}$ $day^{-1}$) | $14.37 \pm 0.37$ |

Presented data were as mean $\pm$ standard deviation ($n$ = 3). All obtained data were analyzed from harvested cells at late exponential phase (LEP).

### 3.4. Fatty Acid Profiles and Biodiesel Characteristics

Table 3 shows the fatty acids (FA) profile of *A. platensis* NIOF17/003 isolate. The highest FA category was saturated SFAs (42.27%), followed by polyunsaturated fatty acids (PUFA) (31.04%), and monounsaturated fatty acids (MUFA) (26.71%). Palmitic acid (C16:0) was the most predominant FA (33.60%), resulting in high SFA content. However, in PUFA category, linoleic acid (C18:2ω6) was the most dominant FA (27.88%), while the rest of PUFA were mainly ω3 fatty acids (3.16%) of alpha-linolenic acid (1.99%, ALA, C18:3ω3), eicosatrienoic acid (0.46%, C20:3ω3, ETE), eicosapentaenoic acid (0.32%, C20:5ω3, EPA), and docosahexaenoic acid (0.36%, C22:6ω3, DHA).

**Table 3.** Fatty acids profile of the isolated *Arthrospira platensis* NIOF17/003.

| Fatty Acids | Percentage (%) of Total FA |
|---|---|
| C6:0 | 0.04 |
| C8:0 | 0.23 |
| C10:0 | 0.13 |
| C11:0 | 0.24 |
| C12:0 | 0.05 |
| C13:0 | 0.08 |
| C14:0 | 1.96 |
| C14:1 | 0.45 |
| C15:0 | 1.75 |
| C15:1 | 0.18 |
| C16:0 | 33.60 |
| C16:1 | 11.34 |
| C17:0 | 1.86 |
| C17:1 | 0.48 |
| C18:0 | 2.33 |
| C18:1 | 14.26 |
| C18:2$\omega$6 | 27.88 |
| C18:3 $\omega$3 | 1.99 |
| C20:3$\omega$3 | 0.46 |
| C20:5 $\omega$3 | 0.32 |
| C22:6 $\omega$3 | 0.39 |
| $\Sigma$STFA | 42.27 |
| $\Sigma$MUFA | 26.71 |
| $\Sigma$PUFA | 31.04 |

SFAs, MUFA and PUFA are saturated, monounsaturated, and polyunsaturated fatty acids.

Table 4 shows the biodiesel characteristics of *A. platensis* NIOF17/003 compared with other species of *Arthrospira*, and marine microalga *Nannochloropsis* as one of the most promising biodiesel species, and those conformed by international standards (ASTM and EN). The values of CN, IV, SV, and CFPP were in conformance with ASTM and EN, while KV was higher recommended by EN and ASTM (Table 4).

**Table 4.** Physicochemical characteristics of biodiesel of *Arthrospira platensis* NIOF17/003 in comparison to different strains of *Arthrospira* and marine microalga *Nannochloropsis*, and the recommended global standards of ASTM and EN biodiesel.

| Strain/Standard | DU | CN | IV | SV | LCSF | CFPP | KV | Db $\geq$ 4 | Reference |
|---|---|---|---|---|---|---|---|---|---|
| *A. platensis* NIOF17/003 | 88.7 | 52.9 | 85.5 | 210.7 | 4.5 | −2.2 | 19.3 | 0.71 | This study |
| *A. platensis* | 12.53 * | 70 | 102 | 191.9 * | 4.28 * | −3 | 12.4 | 0 | [20] |
| *A. platensis* | 99.08 * | 71.02 * | 109.64 * | 110.50 * | 8.35 * | 9.75 * | 4.67 | 0 | [57] |
| *N. oceanica* CCMP531 | 58 | 58 | 66 | 207 * | 5 * | 10 | 4.2 | 5.4 | [58] |
| *N. oceanica* CCNM 1081 | 70–102 * | 36–54 | 88–169 | 195–200 | 4.4–6 | 2.5–2.8 | 4.6 * | 12–36 | [59] |
| *N. oceanica* NIOF15/001 | 57 | 52 | 94 | 200 | 6.7 | 4.6 | 4 | 19.5 | [17] |
| US (ASTM6751−03) [#] | - | $\geq$47 | - | - | - | - | 1.9–6.0 | - | [60] |
| Europe (EN 14214) [#] | - | $\geq$51 | $\leq$120 | - | - | $\leq$5/$\leq$−20 | 3.5–5.0 | - | [61] |

DU is unsaturation degree; CN is cetane number; IV is iodine value (g I2/100 g oil); SV is saponification value (mg KOH g$^{-1}$); LCSF is long-chain saturation factor (wt%); CFPP is cold filter plugging point (°C); KV is kinematic viscosity at 40 °C (mm$^2$ s$^{-1}$), and Db $\geq$ 4 is percent of fatty acids with double bonds equal and/or higher than 4 as % of total fatty acids. * Values were calculated based on the given data using the equations in the references. [#] Values of recommended global standards of USA (ASTM biodiesel), and Europe (EN 14214).

*3.5. Nutritional Value of Biodiesel Byproduct (FL) as Aquaculture Feed*

Population (ind. L$^{-1}$) and number of females carrying eggs (ind. L$^{-1}$) of rotifer *B. plicatilis* were those fed on different concentrations of FL (FL$_{0.2}$, FL$_{0.4}$, FL$_{0.6}$, FL$_{0.8}$, FL$_{1.0}$, respectively) and DW (DW$_{0.2}$, DW$_{0.4}$, DW$_{0.6}$, DW$_{0.8}$, DW$_{1.0}$, respectively) of *A. platensis* NIOF17/003, shown in Figure 4. Among all evaluated treatments, the highest significant rotifer population was obtained when rotifer feed FL$_{0.6}$ concentration (0.6 g of algal-free lipid), followed by FL$_{0.4}$, and FL$_{0.8}$. On the other hand, rotifer feed on FL$_{0.6}$, FL$_{0.8}$, and DW$_{0.4}$ were recorded as the highest significant number of females carrying eggs (Figure 4).

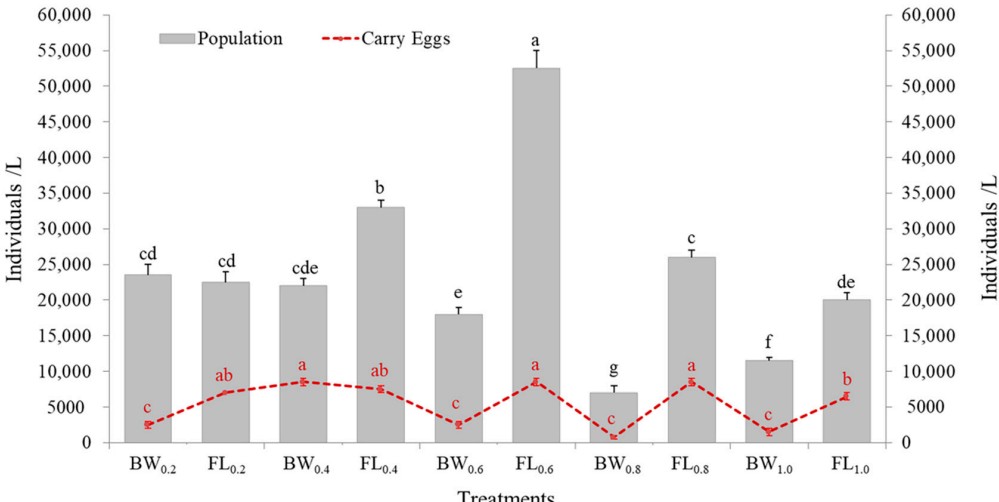

**Figure 4.** Population and number of females carrying eggs of rotifer *B. plicatilis* are those fed on different concentrations of *Arthrospira platensis* NIOF17/003 free lipid (FL, as biodiesel byproduct) in comparison to *A. platensis* dry weight (DW) before lipid extraction. $FL_{0.2}$, $FL_{0.4}$, $FL_{0.6}$, $FL_{0.8}$, $FL_{1.0}$, respectively, and $DW_{0.2}$, $DW_{0.4}$, $DW_{0.6}$, $DW_{0.8}$, $DW_{1.0}$, respectively, are different concentrations (0.2, 0.4, 0.6, 0.8, 1.0 g) of FL or DW of *A. platensis* NIOF17/003. Data presented are mean $\pm$ standard error ($n$ = 3). Different letters in each plotted series indicate significant differences at $p < 0.05$ using Duncan's test.

## 4. Discussion

*Arthrospira*, the super natural food, has a long history of human consumption in Africa back to the 9th century AD. *Arthrospira* cakes, called dihé, are harvested from the Kossorom Lake (Chad Lake) and sun-dried in greenish cakes. Dihé is a blooming of natural *Arthrospira* in the soda lakes of Chad and Niger and considered a common diet component of the population communities of the area. Annually, about 40 t of dihé cakes have been harvested from Chad Lake with local trading value more than US $100,000. Therefore, *Arthrospira* cakes, dihé, represent a significant contribution to the economy of Chad and Niger [62]. *Arthrospira* is the dominant species in alkaline lakes of the Rift Valley and semi-desert Sudan-Sahel zone [63]. As well, *Arthrospira* has been reported as dominant in El-Khadra Lake, the alkaline lakes located in Wadi El-Natrun, Egypt [34–38]. Although *Arthrospira* is a widely distributed species in Africa and consumed by different African communities, achieving significant contributions to the national economy of many African countries, very few studies were conducted on the biotechnological applications of *Arthrospira* along with Africa [64], and the biomass production of *Arthrospira* is still limited [65].

To achieve strong, successful and commercial aqua-biotechnological applications, screening, isolation, molecular identification, and physiological characterizations are needed for native aquatic organisms [8,17,66,67]. In the current study, the native cyanobacteria strain *A. platensis* NIOF17/003 was isolated, in spring 2017, from surface water of the El-Khadra Lake, Wadi El-Natrun, Egypt and deposited in the GenBank database (Accession Number MW396472). *A. platensis* NIOF17/003 showed 52% of protein and 14% of carbohydrate, at LEP based on dry weight percentage (DW%). Our results are in the same line with the results obtained by Marrez et al. [37] who found that the protein and carbohydrate contents of *A. platensis*, isolated from El-Khadra Lake and cultured on different growth media, were from 49.5% to 59.8% and from 12.4% to 22.8%, respectively. Whereas, Aly and Gad [36] reported that *A. platensis*, isolated from El-Khadra Lake and cultured outdoor at semi-pilot scale cultivations, showed 64% protein and 10% carbohydrate. Albert et al. [68] stated that the protein content of *A. platensis* sampled from three production sites, around Chad Lake, were ranged from 50.2% to 58.6%. It is scientifically recognized that *Arthrospira* has a protein content that varies from 50 to 70% of its DW, overriding that of meat, soybeans, eggs, grains, or dried milk [69]. Interestingly, *Arthrospira* has a complete protein that

contains all essential amino acids, which form 47% of total protein weight. According to the literature, leucine, valine, and isoleucine are the highest values for the essential amino acids reported in *Arthrospira*, while methionine and cysteine (sulfur-containing amino acids) are the most poorly essential amino acids [69,70].

The present study found that *A. platensis* NIOF17/003 showed 8.52% of lipid percentage, at LEP, based on DW%. Our results agreed with Aly and Gad [36] who found that *A. platensis*, isolated from El-Khadra Lake and cultured outdoor at semi-pilot scale cultivations, showed 8% of lipid. Moreover, Marrez et al. [37] found that the lipid content of *A. platensis*, isolated from El-Khadra Lake and cultured on different growth media, were ranged from 6.57% to 8.13%. Whereas, Albert et al. [68] reported that the lipid content of *A. platensis*, sampled from three production sites around Chad Lake, was varied from 3.4% to 9.9%. Habib et al. [21] found that the lipid content of *A. platensis* varied from 5.6 to 7%, while Yoshida and Hoshi [71] cited that the lipid content of *A. platensis* may reach to 11%. Matta et al. [72] cited that the lipid content of *A. platensis* ranging from 4 to 16%. However, the differences between our results and those obtained by other studies, even those isolated from the same site of El-Khadra Lake, may be due to different culture systems, scales, extraction methods, environmental conditions [21,36,37,68–73], or available nutrient limitation [73].

The aquatic organisms are a sustainable source of PUFA, MUFA, SFAs, palmitic acid, and fatty acid methyl esters [74,75]. Among all aquatic organisms, microalgal cells are the most sustainable, attractive, and promising source [1–3,17–20,69,72,73]. Cohen [76] stated that about 50 percent of the total lipid of *Arthrospira* is fatty acids. In the current study, *A. platensis* NIOF17/003 isolate showed 3.16% of fatty acids as belonging to omega-3 (mainly ALA, ETE, EPA, and DHA), 31.04% of PUFA, 26.71% of MUFA, 42.27% of TFA, and 33.6% of palmitic acid (C16:0), which presented as the most predominant SFA. These results proved that, besides the high protein contents of *A. platensis* NIOF17/003, this isolate is considered an attractive source for PUFA, omega-3 fatty acids, and palmitic acid (C16:0), which considered the most considered fatty acid for producing biodiesel from microalgae. PUFA and omega-3 fatty acids have significant nutritional value [69], which means that *A. platensis* NIOF17/003 isolate is considered a source for human nutrition, beside its significant quantity of protein. Our results are in the same line with Marrez et al. [37] who found that *A. platensis* grown in different media showed PUFA ranged from 15.86 to 32.07%, SFAs ranged from 67.93 to 84.14%, and palmitic acid (C16:0) ranged from 11.11 to 45.33% of total FAs. Mostafa et al. [20] indicated that *A. platensis*, grown at an indoor scale and enriched with Zarrouk medium, showed 89.6%, 6.2%, and 2.11% of SFAs, MUFA, and PUFA. Habib et al. [21] concluded that PUFA of *A. platensis* may reach up to 30% of SFAs.

Although there is high protein content of *A. platensis*, which varies between 50 and 70% of its DW [69], many authors cited that *A. platensis* is an attractive, sustainable, and promising source for biodiesel production [2,20,23,25–27]. Many authors concluded that the most important keys for successful biodiesel production from microalgae are lipid productivity and biomass productivity [55,77,78]. In spite of low lipid content (8.5%) presented in *A. platensis* NIOF17/003 isolate in the current study, this isolate showed high biomass productivity (143.83 mg $L^{-1}$ day$^{-1}$) and lipid productivity (14.37 mg $L^{-1}$ day$^{-1}$), at LEP, with respect to biodiesel production, although the obtained results gives low relative lipid content compared to green microalgae. *A. platensis* showed a comparable biomass and lipid productivity, which make this isolate an economically viable biodiesel feedstock.

According to the literature, *Nannochloropsis* is considered one of the most promising economic species for production of biodiesel [17,58,59]. In the present work, the reported biomass productivity was 1.9, 5.7, and 7.6 times higher than those reported for *N. oceanica* NIOF 15/001 [17], *N. oceanica* CCNM 1081 [59], and *N. oceanica* CASACC201 [79], respectively. As well, the lipid productivity recorded in the current study was 1.3, 2.1, and 7.1 times higher than those reported for *N. oceanica* CCNM 1081 [59], and *N. oceanica* CASACC201 [79], and *N. oculata* [80], respectively. Moreover, the comparison between the

lipid profile of *Arthrospira* and *Nannochloropsis* is an important point to discuss. About 50% of the total lipid of *Arthrospira* is fatty acids [76], while the composition of *Nannochloropsis* lipid fractions includes hydrocarbons, triacylglycerides (TAGs), free fatty acids (FFAs), amines, sterols, terpenes, and organic acids [81,82].

The optimal selection of microalgae strains for potential biodiesel production requires high lipid productivity as well as requiring acceptable specifications of the FAMEs [17]. Many authors cited that the biodiesel properties are mainly influenced by the fatty acid profile [4,49,54,83,84]. In the current study, the biodiesel properties of *A. platensis* NIOF17/003 isolate, comparing to *Arthrospira, Nannochloropsis*, and the recommended global standards of Europe (EN14214) and US (ASTM6751−03), are presented in Table 4. The values of CN, IV, and CFPP are in correspondence with those conformed by international standards of ASTM6751–03 [60] and EN14214 [61]. As well, the values of DU, SV, and LFPP are in the range of the recommended biodiesel feedstock species of *N. oceanica* NIOF15/001 [17], *N. oceanica* CCMP531 [58], *N. oceanica* CCNM 1081 [59], and *A. platensis* [20,57]. The only observed higher value in the current study was KV (Table 4).

Our results are in accordance with those obtained by Mostafa [20] who evaluated biodiesel properties for *A. platensis*, comparing to ASTM, EN, and Egyptian petro-diesel. On the other hand, $Db \geq 4$ of *A. platensis* NIOF17/003 isolate was lower than those obtained by *Nannochloropsis* (Table 4) and this is due to the high level of PUFA. The current study reported that, regarding biomass productivity, lipid productivity, and characteristics of biodiesel produced from *A. platensis* NIOF17/003 isolate, this isolate is an attractive and promising biodiesel feedstock. Our results are matched with many studies that reported that microalgae, especially *A. platensis*, are promising biodiesel feedstock [2,20,23,25–27,83,84].

On the other hand, aquaculture is a promising industry that supplies essential and cheap animal protein to meet a growing world population [74,85]. Protein is considered the most important ingredients in aquaculture diets. Recently, the microalgae biomass has been extensively utilized in aquatic animal feed, resulting in enhanced growth and survival [9]. Marine rotifer *Brachionus plicatilis* is considered the main zooplankton utilized as live food for marine larvae in marine hatcheries [81,82]. In the present study, different concentrations of FL ($FL_{0.2}$, $FL_{0.4}$, $FL_{0.6}$, $FL_{0.8}$, and $FL_{1.0}$), compared to the same different concentrations of *A. platensis* NIOF17/003 dry weight ($WB_{0.2}$, $WB_{0.4}$, $WB_{0.6}$, $WB_{0.8}$, and $WB_{1.0}$), were evaluated as feed for marine rotifer *B. plicatilis*. Among all treatments, application of 0.6 g of FL of *A. platensis* NIOF17/003 ($FL_{0.6}$ concentration) resulted in the highest rotifer population and the highest significant number of females carrying eggs.

Interestingly, this work is the first study that evaluates *A. platensis* free lipid (FL) as feed for marine rotifer *B. plicatilis*. Our results are in the same line with the many studies that evaluated the effect of microalgal-free lipid as feed for Artemia [17,86,87] and concluded that microalgae-biodiesel byproduct (FL) significantly improved survival and growth of *Artemia franciscana*. Ashour et al. [17] reported that applying 0.1 g $L^{-1}$ of FL of *N. oceanica* NIOF15/001 significantly improved survival (500%) and growth (40%) of *A. franciscana*. Abomohra et al. [86] concluded that FL of *S. obliquus* up to 0.2 g $L^{-1}$ improves the survival of Artemia up to 50%. Moreover, El-Kassas et al. [87] confirmed that a mixed diet of lipid-free algal biomass (0.5 g $L^{-1}$) achieved the highest survival (91.8%) and the highest average of fresh weight (4.2 mg) of Artemia up to about 50%.

## 5. Conclusions

Although *Arthrospira* is a widely distributed species in Africa and consumed by different African communities, achieving significant contributions to the national economy of many African countries, very few studies were conducted on the biotechnological applications of *Arthrospira* along Africa, and the biomass production of *Arthrospira* is still very limited. To achieve strong successful commercial and sustainable aqua-biotechnological microalgae applications, screening, isolation, molecular identification, and physiological characterizations are needed. In the current study, the new cyanobacteria strain *Arthrospira platensis* NIOF17/003 was isolated from the water of El-Khadra Lake, an alkaline lake

located in Wadi El-Natrun, Egypt, and deposited in the GenBank database under accession number MW396472. *A. platensis* NIOF17/003 showed that the values of protein, carbohydrate, and lipid were 52%, 14%, and 8.5%, respectively. Biomass productivity and corresponding lipid productivity was 143.83 mg $L^{-1}$ day$^{-1}$ and 14.37 mg $L^{-1}$ day$^{-1}$ respectively. Biodiesel characteristics of *A. platensis* NIOF17/003 isolate were compatible with those recommended by international standards of US (ASTM6751−03) and Europe (EN14214), as well as different strains of *Arthrospira* and *Nannochloropsis*. Thus, the present work concluded that the isolate *A. platensis* NIOF17/003 is a favorable candidate for biodiesel production, due to its high biomass productivity, lipid productivity, and compatible biodiesel quality standards. Moreover, application of the residual algal-free lipid (at 0.6 g per 20,000 rotifer individuals) significantly increased the rotifer population and the number of females carrying eggs, confirming that FL, rich in protein and carbohydrate, may be efficiently utilized for sustainable rotifer production. Further studies will be conducted on *A. platensis* NIOF17/003 that can be grown easily in photobioreactors (PBR), open ponds, agricultural and industrial wastewater to enhance lipid productivity of *A. platensis* NIOF17/003 to capitalize its feasibility for mass production and biodiesel production.

**Author Contributions:** Conceptualization, M.A.Z., A.M.N., M.A. and A.M.M.H.; methodology, M.A., A.M.M.H., M.M.M. and M.E.E.; software, M.A., M.E.E. and A.E.A.; validation, A.T.M., H.A.H. and A.G.; formal analysis, A.E.A., M.M.M., M.A. and M.E.E.; investigation, M.M.M., M.A., A.T.M., H.A.H. and A.G.; data curation, A.M.M.H., A.E.A., M.M.M., A.T.M. and M.E.E.; writing—original draft preparation, M.A., M.E.E., M.M.M., A.E.A. and A.M.M.H., writing—review and editing, M.A., H.M.K. and M.E.E.; supervision, M.A.Z., A.M.N. and H.M.K.; project administration, M.A.; and funding acquisition, A.T.M., H.A.H. and A.G. All authors have read and agreed to the published version of the manuscript.

**Funding:** The authors appreciated Taif University Researchers Supporting Project number TURSP-2020/39, Taif University, Taif, Saudi Arabia.

**Institutional Review Board Statement:** Not applicable.

**Informed Consent Statement:** Not applicable.

**Data Availability Statement:** Not applicable.

**Acknowledgments:** The authors gratefully acknowledge the Taif University Researchers Supporting Project number TURSP-2020/39, Taif University, Taif, Saudi Arabia for technical and financial support.

**Conflicts of Interest:** The authors declare no conflict of interest.

## Abbreviations

| | |
|---|---|
| ANOVA | analysis of variance |
| CFPP | cold filter plugging point |
| CN | cetane number |
| DU | unsaturation degree |
| DW | dry weight |
| EC | electrical conductivity |
| EEP | early exponential phase |
| FA | fatty acid |
| FAME | fatty acid methylated ester |
| FL | free lipid biomass |
| GC-MS | gas chromatography-mass spectrometry |
| IV | iodine value |
| KV | kinematic viscosity |
| LCSF | long-chain saturation factor |
| LEP | late exponential phase |
| MUFA | monounsaturated fatty acid |
| NIOF | National Institute of Oceanography and Fisheries |

| NJ | neighbor-joining |
| OD | optical density |
| PD | parameter distance |
| PUFA | polyunsaturated fatty acids |
| RT | retention time |
| SD | standard deviation |
| SV | saponification value |
| TSS | total soluble salts |
| WB | whole biomass |

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
