# Peer review of "Potential Applications of Native Cyanobacterium Isolate (Arthrospira platensis NIOF17/003) for Biodiesel Production and Utilization of Its Byproduct in Marine Rotifer (Brachionus plicatilis) Production"

_sustainability, doi:10.3390/su13041769_

Round 1

Reviewer 1 Report

The work on the search for new types of algae for the production of biodiesel is part of the contemporary search for sources of raw materials for fuel production. The publication is very interesting. However, the following should be clarified:

1) Line 179 The text shows that FAME is produced directly in the algae and is extracted. Is transesterification necessary in this situation? In my opinion, oil is formed in algae.

2) Line 187 The equation uses DW and the abbreviations used CDW. Was the same parameter used? 

3) Line 188 - 195 section 2.4. In my opinion first should be described examined parameters, as a second should be described calculated parameters. Now is not clearly which parameters was calculated. Separation of these parameters will help the reader to follow the content of the publication.

4) Line 191-194 These parameters was not determined. Its was calculated.

5) Line 198 If this DW is the same as in 2.3? From the text contained in this line it is not clear that it refers to the dry weight.

6) Table 1 No subscripts were used in the formulas CO3 and HCO3 used in Table 1. 

7) Line 271 In my opinion, it should be added the information that the values contained in table 4 have been calculated or measured.

8) Table 4 In my opinion, errors in the calculated parameters should be added.

9) Line 374 In my opinion, it would be worth emphasizing that the low lipid content is compensated by the rapid growth of algae compared to oilseeds.

Reviewer 2 Report

In the paper presented by Zaki et al. authors presented potential application of native cyanobacterium isolate for biodiesel production and potential application of its byproducts in marine rotifer production. The authors performed isolation, identification and characterisation of selected strain. I find this article well written with all necessary components. The methods and material section is written with all of necessary details. In results and discussion section obtained results are presented in clear way and are easy to fallow and discussion is also informative and all results are compared with previous researches. Some minor mistakes should be corrected. Some of them are:

  1. In the list of authors, is one of author missing? It is written Mostafa E. Elshobary 8 and 9. If author is working in both institutions and should be removed. If one author is missing, name should be added.
  2. Row 51., Overnight should be removed
  3. Row 63. And many should be removed
  4. Row Dis missing at the end of sentence.
  5. Row 127. E in electrical conductivity should be written with small letter.
  6. Row 164. Value of air flow should be added
  7. Row 171. DW should not be in italic
  8. Equation 2. X should be × like it is in other equations in the manuscript. In equation cell dry weight is written DW and in symbol explanation it is CDW. Make it uniform. T for time should be written in small letter
  9. row 203. ml should be mL
  10. rows 225-227. The text is repeating the results from Table 1. It should be removed.
  11. Table 1. CO3 and HCO3, 3 should be in subscript. ds/m should be ds m-1.
  12. row 247. Comma after 0.845 should be removed
  13. row 279. KOH g l-1. Should be L and -1 should be in superscript.
  14. Overall, there are a lot of abbreviations in the manuscript and it would be easier to fallow all if there was a List of abbreviations.
  15. Row 298. n should be in italic.

Author Response

SUMMARY OF AUTHOR(S) RESPONSE TO REVIEWER’S COMMENTS

Manuscript #: sustainability-1097937

Title of the Manuscript: Potential applications of native cyanobacterium isolate (Arthrospira platensis NIOF17/003) for biodiesel production and utilization of its byproduct in marine rotifer (Brachionus plicatilis) production

Author(s): Mohamed A. Zaki, Mohamed Ashour, Ahmed M.M. Heneash, Mohamed M. Mabrouk, Ahmed E. Alprol, Hanan M. Khairy, Abdelaziz M. Nour, Abdallah Tageldein Mansour, Hesham A. Hassanien, Ahmed Gaber, Mostafa E. Elshobary

Reviewer 2# Comment

Author(s) response

Minor Comments:

In the paper presented by Zaki et al. authors presented potential application of native cyanobacterium isolate for biodiesel production and potential application of its byproducts in marine rotifer production. The authors performed isolation, identification and characterisation of selected strain. I find this article well written with all necessary components. The methods and material section is written with all of necessary details. In results and discussion section obtained results are presented in clear way and are easy to fallow and discussion is also informative and all results are compared with previous researches. Some minor mistakes should be corrected. Some of them are:

1.     In the list of authors, is one of author missing? It is written Mostafa E. Elshobary 8 and 9. If author is working in both institutions and should be removed. If one author is missing, name should be added.

No there is no other author and Mostafa E. Elshobary is working in two institutes.

2.     Row 51. Overnight should be removed

The correction has been made

3.     Row 63. And many should be removed

The correction has been made

4.     Row Dis missing at the end of sentence.

The correction has been made

5.     Row 127. E in electrical conductivity should be written with small letter.

The correction has been made

6.     Row 164. Value of air flow should be added

The air flow of (3 cm3 min-1) has been added (please refer to Page 3, line #180)

7.     Row 171. DW should not be in italic

The correction has been made

8.     Equation 2. X should be × like it is in other equations in the manuscript. In equation cell dry weight is written DW and in symbol explanation it is CDW. Make it uniform. T for time should be written in small letter

The correction has been made

9.     Row 203. ml should be mL

The correction has been made

10.  Rows 225-227. The text is repeating the results from Table 1. It should be removed.

The repeated sentences have been deleted.

11.  Table 1. CO3 and HCO3, 3 should be in subscript. ds/m should be ds m-1.

The correction has been made

12.  Row 247. Comma after 0.845 should be removed

The correction has been made

13.  Row 279. KOH g l-1. Should be L and -1 should be in superscript.

The correction has been made

14.  Overall, there are a lot of abbreviations in the manuscript and it would be easier to fallow all if there was a List of abbreviations.

The journal File Format did not include this part, however, if acceptable for the journal file format, the List of abbreviations has been added after abstract section

15.  Row 298. n should be in italic.

The correction has been made

We would like to extend our sincere thanks and appreciation to the reviewers and editorial board. In fact, their comments and guidance added a lot to the research and increased its scientific content. Therefore, the words cannot express their gratitude for their time and effort they put in evaluating this research.

Reviewer 3 Report

This manuscript illustrates very well focus on investigating the Potential applications of native cyanobacterium isolate for biodiesel production. The authors showed a good description on the analysis methods and research results of this study, and I recommend revisions according to the following comment:

  1. The manuscript is adequately presented in a very good scientific manner. 
  2. It is thoroughly researched. 
  3. Among current synthesis methods, transesterification is the main industrial production method of biodiesel now. So the biodiesel can be made from vegetable oil, animal fat and waste cooking oil by transesterification. In this study, biodiesel can be obtained from algal biomass, algae biodiesel has great application potential and economic benefits. But at present, many researchers have reported the method of preparing biodiesel from algae, so the authors should better highlight their key findings in comparison with other researchers.

Author Response

SUMMARY OF AUTHOR(S) RESPONSE TO REVIEWER’S COMMENTS

Manuscript #: sustainability-1097937

Title of the Manuscript: Potential applications of native cyanobacterium isolate (Arthrospira platensis NIOF17/003) for biodiesel production and utilization of its byproduct in marine rotifer (Brachionus plicatilis) production

Author(s): Mohamed A. Zaki, Mohamed Ashour, Ahmed M.M. Heneash, Mohamed M. Mabrouk, Ahmed E. Alprol, Hanan M. Khairy, Abdelaziz M. Nour, Abdallah Tageldein Mansour, Hesham A. Hassanien, Ahmed Gaber, Mostafa E. Elshobary

Reviewer 3# Comment

Author(s) response

Minor Comments:

This manuscript illustrates very well focus on investigating the Potential applications of native cyanobacterium isolate for biodiesel production. The authors showed a good description on the analysis methods and research results of this study, and I recommend revisions according to the following comment:

1.     The manuscript is adequately presented in a very good scientific manner. 

We would like to thank you and the reviewer for evaluating the manuscript with these constructive comments. FAMEs preparation has been added in details (please refer to Page 4, line #197-203)

2.     It is thoroughly researched. 

3.     Among current synthesis methods, transesterification is the main industrial production method of biodiesel now. So the biodiesel can be made from vegetable oil, animal fat and waste cooking oil by transesterification. In this study, biodiesel can be obtained from algal biomass, algae biodiesel has great application potential and economic benefits. But at present, many researchers have reported the method of preparing biodiesel from algae, so the authors should better highlight their key findings in comparison with other researchers.

We would like to extend our sincere thanks and appreciation to the reviewers and editorial board. In fact, their comments and guidance added a lot to the research and increased its scientific content. Therefore, the words cannot express their gratitude for their time and effort they put in evaluating this research.
